# From miRNA Target Gene Network to miRNA Function: miR-375 Might Regulate Apoptosis and Actin Dynamics in the Heart Muscle via Rho-GTPases-Dependent Pathways

**DOI:** 10.3390/ijms21249670

**Published:** 2020-12-18

**Authors:** German Osmak, Ivan Kiselev, Natalia Baulina, Olga Favorova

**Affiliations:** 1National Medical Research Center for Cardiology, 121552 Moscow, Russia; kiselev.ivan.1991@gmail.com (I.K.); tasha.baulina@gmail.com (N.B.); olga.favorova@gmail.com (O.F.); 2Pirogov Russian National Research Medical University, 117997 Moscow, Russia

**Keywords:** network analysis, microRNA, miR-375, myocardium, heart, miRNA function, Rho-GTPases

## Abstract

MicroRNAs (miRNAs) are short, single-stranded, non-coding ribonucleic acid (RNA) molecules, which are involved in the regulation of main biological processes, such as apoptosis or cell proliferation and differentiation, through sequence-specific interaction with target mRNAs. In this study, we propose a workflow for predicting miRNAs function by analyzing the structure of the network of their target genes. This workflow was applied to study the functional role of miR-375 in the heart muscle (myocardium), since this miRNA was previously shown to be associated with heart diseases, and data on its function in the myocardium are mostly unclear. We identified *PIK3CA, RHOA, MAPK3, PAFAH1B1, CTNNB1, MYC, PRKCA, ERBB2,* and *CDC42* as key genes in the miR-375 regulated network and predicted the possible function of miR-375 in the heart muscle, consisting mainly in the regulation of the Rho-GTPases-dependent signaling pathways. We implemented our algorithm for miRNA function prediction into a Python module, which is available at GitHub.

## 1. Introduction

MicroRNAs (miRNAs) are a class of short, single-stranded, non-coding ribonucleic acid (RNA) molecules, about 21–25 nucleotides in length, which regulate gene expression at the post-transcriptional level, through sequence-specific interaction with mRNAs of target genes followed by their degradation or repression of translation [1]. MiRNAs are involved in the regulation of main biological processes, such as apoptosis or cell proliferation and differentiation [2]. In the cardiovascular system, miRNAs play an essential role in the growth and development of cardiomyocytes, contractility, heart rhythm control, angiogenesis, and metabolism of lipids [2,3,4].

We previously found the association of circulating miR-375 level with myocardial infarction (MI) [5]. The study [6] showed the possible functional role of this miRNA in the heart muscle, through the regulation of the PI3K/Akt signaling pathway involved in many biological processes, in particular, cell proliferation and differentiation. Excluding the aforementioned study, data on the miR-375 function in the heart muscle (myocardium) are lacking.

In this study, we investigated the possible function of miR-375 in the heart muscle by analyzing the structure of the network of tissue-specific target gene interactions.

## 2. Results

Using the Human Protein Atlas database, we selected a total of 7944 protein-coding genes, expressed in the heart muscle. In this tissue, according to the MiRTarBase database, there were 267 target genes of miR-375 (about 3.5% of the total number of protein-coding genes). To evaluate the possible functional role of miR-375 in the heart muscle, we analyzed the structure of the gene–gene interaction network, composed of these 267 genes. Of them, 82 genes are indexed in the String database and have at least one connection. The largest connected component (LCC) of the network contains 53 genes (Figure 1A). Other connected components contained less than six genes each. If extracting random gene sets equal in cardinality to the considered sets of miR-375 target genes, the probability of getting LCC of the same sizes is less than 0.0001, which corresponds to the level of statistical significance *p* < 0.05.

To identify the key genes of the miR-375 regulated network, we extracted top genes successively, based on the centrality of their nodes from the LCC shown in Figure 1A, until the LCC cardinality hit a plateau (Figure 1B). The extracted key miR-375 target genes, essential for the connectivity of the LCC, were (in descending order of centrality): *PIK3CA, RHOA, MAPK3, PAFAH1B1, CTNNB1, MYC, PRKCA, ERBB2,* and *CDC42* (Figure 1C).

To investigate the main functions of the key miR-375 target genes, we performed their overrepresentation analysis in Reactome signaling pathway gene sets. In total, the key genes are significantly overrepresented (*p*-value < 0.05) in 94 signaling pathways (Appendix A).

The considered genes are primarily involved in various signal transduction pathways. Many of the genes of high centrality encode transcription factors, located at the very end of signal transduction pathways bearing the corresponding names, particularly RHO-, MAPK-, PI3K/AKT-, MYC- and ERBB2-signaling pathways.

Furthermore, we ranked the overrepresented Reactome pathways by the number of key miR-375 target genes and filtered out those that included less than three of the key genes. This allowed us to exclude from consideration side pathways and to focus only on pathways that are to a greater extent represented by key miR-375 target genes (Figure 2).

As can be seen from Figure 2, signaling pathways containing key miR-375 target genes are divided into two large clusters. The first cluster mainly includes the Rho-, NTRK-, ERBB2-, PI3K/AKT-signaling pathways (shown as the red branches of the dendrogram); the second - TGFb-, WNT- and SMAD-signaling pathways (shown as the green branches of the dendrogram). Nineteen of the 22 sets from the first cluster are directly involved in Rho-kinase signaling, and/or include at least one gene from this pathway (*RHOA* or *CDC42*); the exceptions are three pathways, linked to ERBB2- and SCF-KIT-signaling. Ten sets with high mean centrality from the first cluster also include the *PIK3CA* gene. All Reactome sets from the second cluster include the *MYC* gene, and all sets with high mean centrality among them also include the *RHOA* gene. Thus, most of the signaling pathways from both clusters of the miR-375 key target genes are associated with Rho-GTPases-dependent signal transduction cascades.

## 3. Discussion

In this study, we analyzed gene–gene interaction networks of miR-375 target genes expressed in the heart muscle, based on data of gene expression of the protein-coding genes in this tissue from the Human Protein Atlas database. Results of the statistical overrepresentation analysis demonstrate that key target genes of miR-375 are involved in intracellular signal transduction, predominantly through Rho-GTPase-signaling. PI3K/AKT-, MAPK-, WNT-, NTRK-, ERBB2-, and EGFR- signaling pathways were also overrepresented with key miR-375 target genes. However, PI3K/AKT-, MAPK- pathways are known to be significantly intermingled with Rho-GTPase signaling [7,8], whereas WNT-, NTRK-, ERBB2-, and EGFR-pathways may function via Rho-GTPase dependent transducers [9,10,11,12,13]. Thus, we may assume that miR-375 action in the heart muscle is primarily directed to the regulation of Rho-GTPase genes’ expression.

Rho-GTPases represent a distinct family within the superfamily of Ras-related small GTPases. This family includes small (about 21 kDa) proteins that function as binary switches involved in the regulation of main cellular processes such as morphogenesis, polarity, movement, and cell division [14]. *CDC42* and *RHOA* encode proteins that, along with Rac1, are the most studied members of the Rho-GTPase family. They induce rearrangements of the actin cytoskeleton and regulate the contractility of actomyosin, and the dynamics of microtubules [15,16].

The role of small GTPases in the development of cardiovascular diseases is described [17]. In animal model of ischemia-reperfusion injury, inhibitors of RhoA/ROCK signaling pathways demonstrate a cardioprotective effect, which is realized through the activation of MAPK and NFkB signaling pathways and a reduction in Bcl-2 levels, leading to a decrease in the apoptotic activity of cells [18,19,20,21,22].

In this work, analyzing the structure of the network of miR-375 target genes, we showed that its function in the heart muscle may be mainly to regulate Rho-GTPases-dependent signaling pathways. Indeed, *CDC42* and *RHOA* genes, which are the direct targets of miR-375, occupy central positions in interaction networks (see Figure 1).

In our analysis, we focused on the gene expression profile in the heart muscle, which includes 7994 protein-coding genes. Among them, 387 genes are most abundant in the heart, compared to other tissues, but only 29 of them are heart-specific, according to the Human Proteome Atlas [23]. Notably, only nine miR-375 experimentally confirmed target genes, namely *LDHB, CFL2, DOK7, NPPB, ANKRD1, NCAM1, TNNI3, C15orf41*, and *B3GALNT1,* were found to be among the most abundant in heart genes; all these genes are non-heart-specific. In addition, the genes listed above were not indicated as key miR-375 targets, according to our bioinformatics analysis. The genes most abundant in the heart are not involved in Rho-GTPase dependent signaling and are not mentioned in studies [6,24] investigating the mechanisms of miR-375 functioning in the heart. Thus, we believe that the predicted function of miR-375 is not specific to the heart muscle. However, additional analysis is needed to fully clarify this.

Recent data demonstrate the protective effect of miR-375 overexpression in cardiomyocytes, after hypoxic-reoxygenation injury [24]. Furthermore, miR-375 downregulation also showed a cardioprotective effect, albeit in another experimental model; in mice with myocardial infarction (MI), caused by ligation of the left anterior descending artery without reperfusion [6]. This contradiction might be explained, based on our data, by taking into account the broad variety of molecular functions of Rho-GTPases in the cell. Inhibition by miR-375 of those Rho-GTPases which are apoptosis activators [25] may have a positive effect in ischemia-reperfusion injury of cultured myoblasts, previously demonstrated in [24]. Likewise, miR-375-dependent inhibition of GTPase Cdc42, one of the key intracellular messengers in angiogenesis [26], may have a negative effect on MI recovery, earlier observed in the model mice [6].

To verify the workflow, we performed an analysis of the miR-375 function in a mouse heart. However, the analysis was restricted by data from the miRTarBase database; there are only 20 experimentally confirmed targets of mmu-miR-375, of which only nine are expressed in the heart tissue, according to the FANTOM5 project (https://www.ebi.ac.uk/gxa/experiments/E-MTAB-3578/Results). According to the String database, these genes do not have connections with each other, which makes the available data insufficient to predict the miR-375 function in the heart muscle of a mouse. None of these genes are Rho-GTPase family members, or miR-375 targets studied in [24], which makes it difficult to extrapolate results obtained in humans and mice, altogether underscoring the need in future for further experimental studies.

In general, this study proposes a workflow for predicting miRNA function in different tissues and cells by analyzing the network structure of its target genes. This approach allowed us to predict the possible function of miR-375 in the heart muscle, realized via the regulation of the expression of Rho-GTPase effectors.

## 4. Limitations and Future Direction

The predicted function of miR-375 certainly needs experimental verification. The use of binary estimates of miRNA target gene expression (expressed/not expressed) in tissue could lead to the loss of some information.

For further research, the question on the specificity of miR-375 functioning as one of the regulators of Rho-GTPases is of interest. Our in silico analysis indicates that this regulation is likely to not be tissue-specific.

## 5. Methods

### 5.1. Workflow

The workflow of the study design is illustrated in Figure 3. The Human Protein Atlas database (https://www.proteinatlas.org/) was used to extract data on gene expression of detected protein-coding genes in the heart muscle [23]. The MiRTarBase database (http://mirtarbase.cuhk.edu.cn/) [27] was used to select direct target genes of miR-375. The STRING database (https://string-db.org/) [28] was used to find molecular interactions between protein products of these genes (only interactions with a high confidence score (score > 0.9) were selected for further analysis). By combining all these data, we constructed the interaction network of miR-375 target genes expressed in the heart tissue. To identify key genes, the LCC was extracted from the network; then top genes were successively extracted by the centrality of their nodes from the LCC, until the LCC disintegrated and lost its connectivity (described below in Section 5.3). Next, we analyzed the overrepresentation of key genes in the Reactome sets. The signaling pathways most overrepresented with key genes were selected, based on the centrality weighted dendrogram clusterization, which was built using data on the centrality of genes in the LCCs.

### 5.2. Network Construction and Analysing

The gene–gene interaction network was constructed using the NetworkX 2.0 package for Python [29]. Target genes served as nodes of the constructed networks, whereas molecular interactions of their protein products served as edges. Splice variants of one gene were combined into one node. Since direct targets of miRNAs frequently behave as network hubs or hubs-bottlenecks [30], the centrality of nodes was calculated as an additive sum of their betweenness and degree centralities; it was implemented through betweenness_centrality() and degree_centrality() functions in Python package NetworkX. The resulting networks and their characteristics were visualized using Cytoscape software [31].

### 5.3. Key Nodes Selection

For key nodes selection, we ordered all of the nodes in decreasing order of their centrality and construct function:(1)$$f\left(n\right):n\to \left|LCC_{n}\right|,\;n\leq N_{LCC},\;n=i$$
where: *n*—number of top centrality nodes removed,
*i*—index of nodes$N_{LCC}$—number of nodes in largest connected component (LCC)$\left|LCC_{n}\right|$—cardinality of the LCC from the network with removed *n* top centrality nodes

Next, we found the minimal *n* when the derivative of this function is equal to zero:(2)$$n_{min}=min\left(n\right),\;n\in \left\{n\;|\;\frac{d\left(f\left(n\right)\right)}{dn}=0\;\right\}$$
(3)$$f\left(n_{min}\right)<N_{LCC}/2,$$

If then
(4)$$key\;nodes\;=\;\left\{nodes_{i}\;|\;i\leq n_{min}\right\}$$
key nodes is a subset of a set of centrally ordered nodes such that the index of the nodes of this subset does not exceed the $n_{min}$.

If $f\left(n_{min}\right)>N_{LCC}/2$, then we took next
(5)$$n\in \left\{n\;|\;\frac{d\left(f\left(n\right)\right)}{dn}=0,\;n>n_{min}\;\right\}$$
and repeated the procedure.

### 5.4. Reactome Gene Set Enrichment Analysis

For gene set enrichment analysis, we used a Reactome API, which was implemented in the Reactome2py v.1.0.1 package for Python3 [32].

Gene set clusterization and dendrogram visualization was performed based on the method described in [33].

### 5.5. Availability of Data and Materials

The code is available at GitHub: https://github.com/GJOsmak/miRNET.

## Figures and Tables

**Figure 1 ijms-21-09670-f001:**
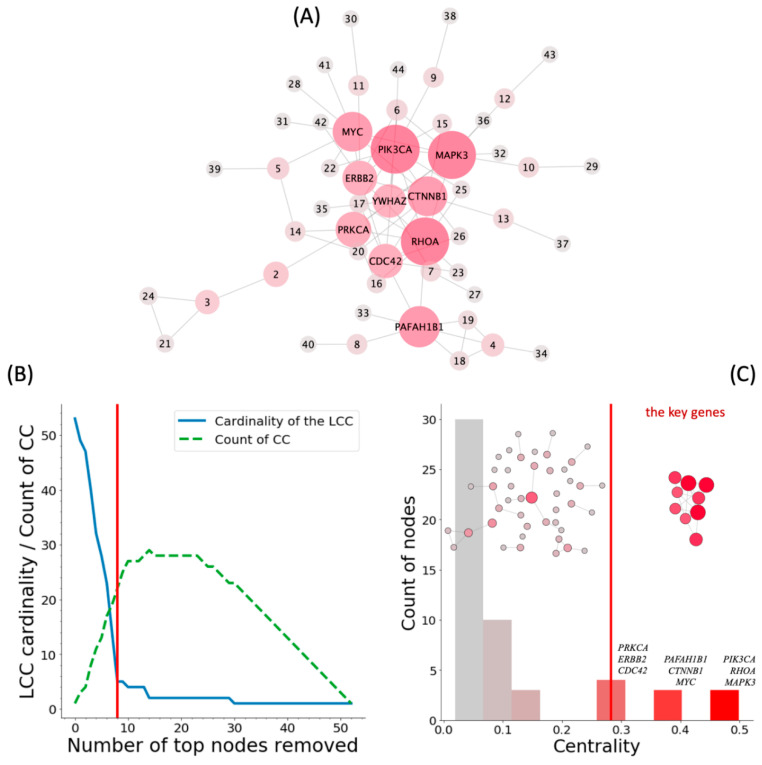
Selection of key protein-coding miR-375 target genes expressed in the heart muscle. (**A**). The largest connected component (LCC) of the network of miR-375 protein-coding target genes, which are fully presented in the Appendix A; the figure provides names only for several genes of the highest centrality. (**B**). After the successive removal of the genes of the highest centrality, the LCC disintegrates and loses its connectivity. The plot shows the reduction in the LCC cardinality (blue line) and simultaneous enlargement of the connected components (CC) number (green dashed line) during nodes’ removal. (**C**) The histogram of centrality distribution of genes in the LCC; right network includes key nodes (marked on the histogram), left network—all but key nodes. The vertical red lines on figures (B) and (C) mark the cut-off level for selection of the key genes when the LCC cardinality hits a plateau. Differences in the color of nodes (A, C) and columns (C) from gray to red mark the increase in the centrality of genes.

**Figure 2 ijms-21-09670-f002:**
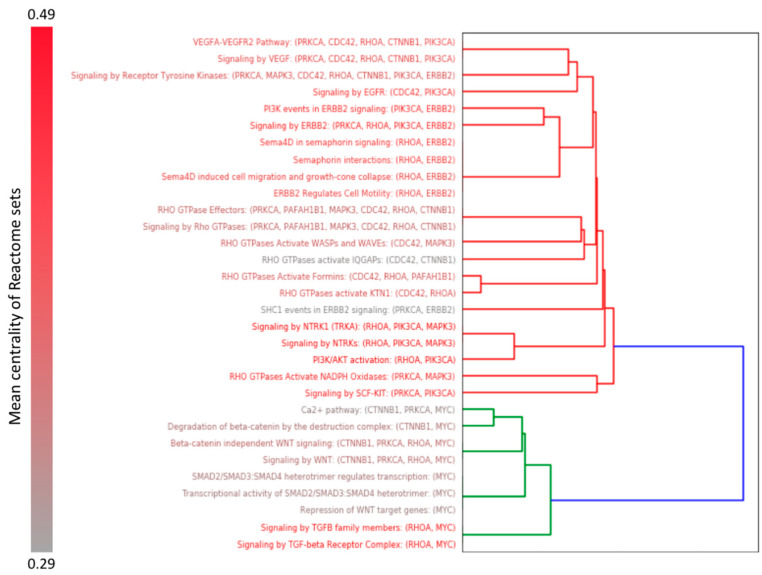
The dendrogram shows clusterization of Reactome pathways, in which the key miR-375 target genes are significantly overrepresented in the heart muscle. The clusters are shown in different colors. The red to gray gradient of text color (shown on the left) reflects the increase in the mean centrality of genes in sets.

**Figure 3 ijms-21-09670-f003:**
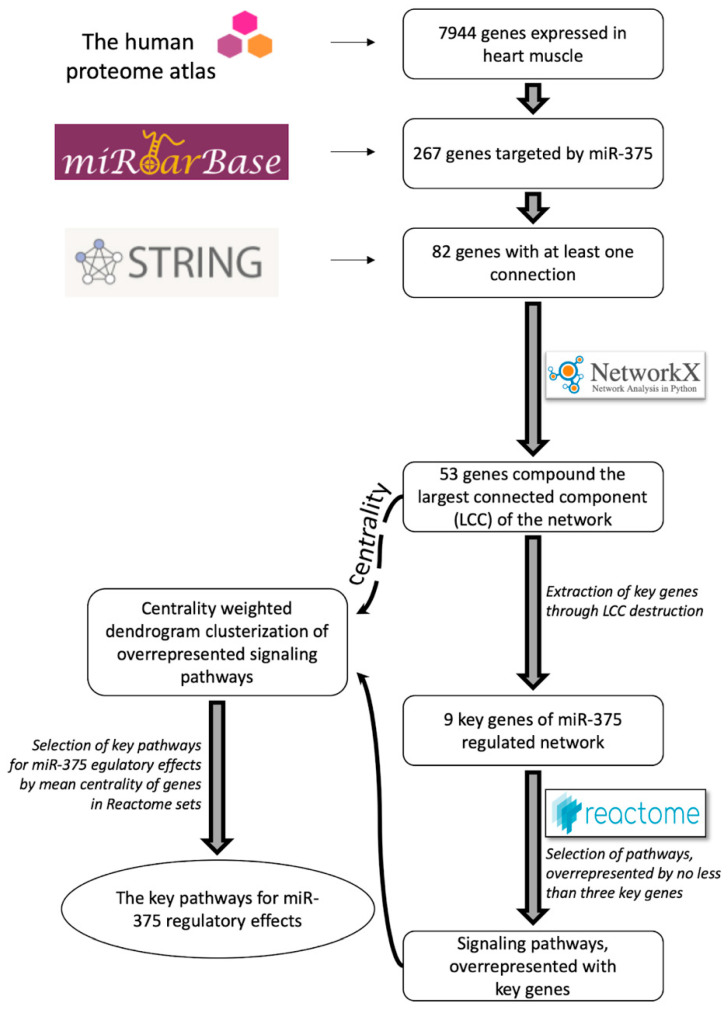
A schematic workflow of prediction of miR-375 function in heart muscle.

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
