# Peer review of "From miRNA Target Gene Network to miRNA Function: miR-375 Might Regulate Apoptosis and Actin Dynamics in the Heart Muscle via Rho-GTPases-Dependent Pathways"

_ijms, 2020, doi:10.3390/ijms21249670_

Round 1

Reviewer 1 Report

In this manuscript Osmak et al propose a workflow for predicting miRNAs function by analyzing the structure of the network of their targets. In my opinion the analyses is very limited and do not provide any relevant information.

Author Response

Dear Reviewer,

We are very sorry that our manuscript could not catch your interest. Indeed, the proposed analysis has limitations, which we describe in the “Limitations and future direction” section. However, we cannot agree that our analysis does not provide any relevant information, since the function of miR-375 in the heart tissue still remains unclear. Relying on the results of the presented study, we propose a workflow to predict this function, which finds additional indirect confirmation in the literature.

Reviewer 2 Report

The article authored by Dr Osmak et al aims to investigate the possible function of miR-375 in the heart muscle by analyzing the structure of the network of tissue-specific target gene interactions. Among the 7944 protein-coding genes expressed in heart muscle initially selected, 267 proved to be targets of miR-375, out of them 82 have at least one connection. From this point, a subset of key genes was defined based on their essential role in the connectivity of the network. These set of genes belongs to two clusters, involving Rho, tyrosine kinase receptor/ PI3K/AKT pathways on the one hand, TGF-B, Wnt and SMAD pathways on the other. Based on these results, the Authors propose that miR-375 might regulate apoptosis and actin dynamics in cardiomyocytes via Rho-GTPases. The paper is well written, and this in silico analysis propose interesting clues concerning the role of miR-375 in cardiomyocytes under pathophysiological conditions. This work continued previous investigations of the Authors concerning miRNA profiling in myocardial infarction patients. One can only regret that the predictions of this in silico analysis were not further validated by an experimental approach, e.g.  the Authors could have performed a PCR array or an antibody array after overexpression or depletion of miR-375 in cardiomyocytes to confirm the involvement of this miRNA on RhoGTPases pathways, or evaluate the functional impact in term of actin remodeling and/or apoptosis. This can be especially important in view of contradictory reports concerning cardioprotective effects of miR-375 overexpression versus downregulation mentioned by Authors in Discussion Section.

A major criticism of the manuscript concerns the quality of the Figures.

The figures are unreadable even with the maximal magnification of the PDF files, and should be redrawn, especially Figures 1 & 2 which are too small. In Figure 1, the black font on the red fill color is not appropriate. Providing a Table could be useful especially for understanding Figure 1A. The Figure 2 does not add value if presented information cannot be visualized and analyzed.

The Authors could provide a workflow chart of their screening of the key target genes of miR-375.

Discussion Section: The fact that Rho GTPases signaling is upstream of PI3K/Akt and MAPK pathways is an over-simplification. For instance, PI3K triggers the activation of a series of Guanine nucleotide Exchange Factors (GEF) with PH domain (e.g. TIAM1, VAV, PREX) that activate Rho, Rac1 and CDC42; ERK1 and AKT phosphorylate and inhibit RAC1.

Author Response

Dear Reviewer,

Thank you for the appreciation of our manuscript and for the recommendations of high significance. We do understand that our in silico analysis needs to be experimentally validated and we reflected this limitation in the submitted manuscript in the “Limitations and future direction” section. Unfortunately, this year we experienced logistical difficulties with performing miR-375 overexpression and depletion experiments due to an epidemiological situation. We hope to continue our experimental work as soon as possible and report the data on transcriptome array and apoptosis experiments. We now stated this limitation in more details and moved it to the end of the Discussion section.

Please, find below the responses to your other comments. All changes, according to your comments, are highlighted in yellow in the manuscript.

Point 1:

A major criticism of the manuscript concerns the quality of the Figures.

The figures are unreadable even with the maximal magnification of the PDF files, and should be redrawn, especially Figures 1 & 2 which are too small. In Figure 1, the black font on the red fill color is not appropriate. Providing a Table could be useful especially for understanding Figure 1A. The Figure 2 does not add value if presented information cannot be visualized and analyzed.

Response 1:

Thank you for pointing out this defect, which was also raised by another Reviewer. We now modified Figure 1 and performed it in more legible format; the colors of nodes now are more transparent. Following your recommendations, we added a new table in the Supplementary material (Table S1), which contains information on the nodes and its centrality in the largest connected component of the miR-375 target gene interaction network. 

We cannot but agree with your comment on Figure 2. We replaced it with a new table (Table S2), which presents data on Reactome signalling pathways with listed overrepresented miR-375 key target genes (p<0.05). 

Point 2:

The Authors could provide a workflow chart of their screening of the key target genes of miR-375.

Response 2:

Thank you for this useful recommendation. We now added a workflow chart of predicting miR-375 function in heart muscle in the “Methods” section (p. 6).

Point 3:

Discussion Section: The fact that Rho GTPases signaling is upstream of PI3K/Akt and MAPK pathways is an over-simplification. For instance, PI3K triggers the activation of a series of Guanine nucleotide Exchange Factors (GEF) with PH domain (e.g. TIAM1, VAV, PREX) that activate Rho, Rac1 and CDC42; ERK1 and AKT phosphorylate and inhibit RAC1.

Response 3:

Following your remark we modified the “Results” and “Discussion” sections (p. 3-4). Now it sounds as:

  • in “Results” section (lines 87-109): “Nineteen out of 22 sets from the first cluster are directly involved in Rho kinase signaling and/or include at least one gene from this pathway (RHOA or CDC42); the exception is three pathways, linked to ERBB2- and SCF-KIT-signaling. Ten sets with high mean centrality from the first cluster also include the PIK3CA gene. All reactome sets from the second cluster include MYC gene, and all sets with high mean centrality among them also include RHOA gene”.
  • in “Discussion” section  (lines  116-120):  “However, PI3K/AKT-, MAPK- pathways are known to be significantly intermingled with Rho-GTPase signaling [7,8], whereas WNT-, NTRK-, ERBB2-, and EGFR-pathways may function via Rho-GTPase dependent transducers [9–13]. Thus, we may assume that miR-375 action in the heart muscle is primarily directed to the regulation of Rho-GTPase genes’ expression”.

Reviewer 3 Report

The present study proposed a workflow using some database for predicting miRNAs function and the authors applied this workflow to study the functional role of miR-375 in the heart muscle. The authors identified some gene such as ERBB2, and CDC42 as key genes in the miR-375 regulated network and predicted the possible function of miR-375 in the heart muscle, consisting mainly in the regulation of the Rho-GTPases-dependent signalling pathways. Although I think that topic of this study is interesting, there are several flaws in this study.

Comments:

  1. The authors focused on the heart muscle, and showed that the functional role of miR-375 in the heart muscle. However, it remains unknown whether the predicated functional role of miR-375 is specific to heart muscle. I wonder that the role of miR-375 is general function. Is it specific to heart muscle? In the first step, the authors selected 7994 protein-coding genes. Does an analysis by focusing on the specific or elevated genes in heart muscle result in same prediction?   
  2. Similar to the authors’ previous report, Reference 21 showed the association of miR-375 with cardiomyocyte injury in mouse model. To confirm the verification of the proposed workflow, the authors also have to perform the workflow by using mouse data, not human data.
  3. The URL of used database should be indicated.
  4. The quality of Figure such as size and resolution is too poor. The authors have to revise them.
  5.  The authors have to carefully check the manuscript. In addition, the authors have to unify the name of journal of references (i.e., abbreviation or full name).

Author Response

Dear Reviewer, 

Thank you for the careful and thoughtful analysis of our manuscript and for the remarks of essential significance. We are sincerely grateful for all your advices. Please, find below the responses to each of your comments, step by step.  All changes in the revised manuscript made in accordance with your recommendations are marked in grey.

Point 1:

The authors focused on the heart muscle, and showed that the functional role of miR-375 in the heart muscle. However, it remains unknown whether the predicated functional role of miR-375 is specific to heart muscle. I wonder that the role of miR-375 is general function. Is it specific to heart muscle? In the first step, the authors selected 7994 protein-coding genes. Does an analysis by focusing on the specific or elevated genes in heart muscle result in same prediction?

Response 1:

Thank you for raising such an important issue. To clarify this point we made the following changes in the submitted manuscript:

- We added a paragraph in “Discussion” section (p. 4, lines: 136-146):

“In our analysis we focused on the gene expression profile in the heart muscle which includes 7,994 protein-coding genes. Among them 387 genes are the most abundant in the heart as compared to other tissues, but only 29 of them are heart-specific according to Human Proteome Atlas. Notably, nine miR-375 experimentally confirmed target genes, namely LDHB, CFL2, DOK7, NPPB, ANKRD1, NCAM1, TNNI3, C15orf41, and B3GALNT1 were found among the genes most abundant in the heart; all these genes are non heart-specific. Besides, listed above genes were not identified as key miR-375 targets according to our bioinformatic analysis. The most abundant in the heart genes are not involved in Rho-GTPase dependent signaling and are not mentioned in studies [6, 24] investigating the mechanisms of miR-375 functioning in the heart. Thus, we believe that the predicted function of miR-375 is not specific to heart muscle, however, additional analysis is needed to fully clarify it”.

-  We also wrote in “Limitations and future direction” section (p. 5, lines 188-190):

“For further research, the question on the specificity of miR-375 functioning as one of the regulators of Rho-GTPases is of interest. Our in silico analysis indicates that this regulation is likely not tissue-specific.”

Point 2:

Similar to the authors’ previous report, Reference 21 showed the association of miR-375 with cardiomyocyte injury in mouse model. To confirm the verification of the proposed workflow, the authors also have to perform the workflow by using mouse data, not human data.

Response 2:

We added corresponding paragraph to the Discussion section (p. 5, lines 171-179):  

“To verify the workflow we performed the analysis of the miR-375 function in mice heart. However, the analysis was restricted by data from miRTarBase database: there are only 20 experimentally confirmed targets of mmu-miR-375, of which only nine are expressed in the heart tissue according to FANTOM5 project (https://www.ebi.ac.uk/gxa/experiments/E-MTAB-3578/Results). According to the String database, these genes do not have connections with each other, which makes the available data insufficient to predict the miR-375 function in mouse heart muscle. None of these genes are Rho-GTPase family members or  miR-375 targets studied in [21] that makes it difficult to extrapolate results obtained in humans and mice, altogether underscoring the need in future experimental studies.”

Point 3:

The URL of used database should be indicated.

Response 3:

Thank you for the comment. We added the corresponding URL (lines 195-197) of used databases in the text. 

Point 4:

The quality of Figure such as size and resolution is too poor. The authors have to revise them.

Response 4:

Thank you for pointing out this defect, which was also raised by another Reviewer. We now modified Figure 1 and performed it in more legible format; the colors of nodes now are more transparent. We added a new table in the Supplementary material (Table S1), which contains information on the nodes and its centrality in the largest connected component of the miR-375 target gene interaction network. We replaced Figure 2 in submitted manuscript with a new table (Table S2), which presents data on Reactome signalling pathways with listed overrepresented miR-375 key target genes (p<0.05).  

Point 5:

The authors have to carefully check the manuscript. In addition, the authors have to unify the name of journal of references (i.e., abbreviation or full name).

Response 5:

Thank you for the remarks. We carefully checked the manuscript and unified the journal titles in References. 

Round 2

Reviewer 1 Report

As the author have mentioned in their reply, the analysis in only predictive and they do not provide any additional information to validate the model.

Reviewer 2 Report

The Authors have taken into account all my recommendations. I consider that the manuscript is acceptable in its present form.

Reviewer 3 Report

The authors have responded to all of my comments. I am satisfied with the authors’ response.